# Efficacy and safety of sacubitril-valsartan in Maintenance Hemodialysis patients with hypertension:A retrospective study

Lili Jiang[1,2], Jing Ran[2], Yu Zhu[2], Lupin Pan[2], Bayi Yang[2], Xue Ran[2], Ying Ran[2], Hejun Ding[1], Jurong Yang[1]*, Shaofa Wu[2]*

1 Department of Nephrology, The Third Affiliated Hospital of Chongqing Medical University, Chongqing, China, 2 Department of Nephrology, Youyang Hospital, a Branch of The First Affiliated Hospital of Chongqing Medical University, China.

* 650230@hospital.cqmu.edu.cn (JY); shaofawu@126.com (SW)

## Abstract

### Background

Hypertension is associated with elevated mortality rates among individuals on maintenance hemodialysis (MHD). However, there is limited information regarding the efficacy and safety of sacubitril-valsartan (SV) in MHD patients suffering from hypertension. Consequently, this study aims to evaluate the efficacy and safety of SV in this specific patient population.

### Methods

We retrospectively reviewed the data of MHD patients with hypertension who were receiving SV treatment at our hospital from January 1, 2023 to June 30, 2024. The SV dose ranged from 50 mg twice daily to 200 mg twice daily. Blood pressure measurements were compared at baseline and during follow-up intervals to assess therapeutic efficacy. Safety evaluations included comprehensive monitoring of treatment-emergent adverse events throughout the observation period.

### Results

A total of 64 MHD patients hypertension (mean age = 54.11 years; male proportion = 73.4%) were ultimately included in the study. After three months of treatment with SV, the mean reductions in systolic blood pressure, diastolic blood pressure, and pulse pressure were 21.06 ± 15.82 mmHg (P < 0.001), 8.41 ± 13.09 mmHg (P < 0.001), and 12.65 ± 13.17 mmHg (P < 0.001), respectively. With respect to safety, new-onset intradialytic hypotension was observed in three patients. The study also included twelve patients with hyperkalemia. Importantly, no patients reported experiencing cough, angioedema, or abnormal liver biochemistry. Among the 64 subjects, none required a dose reduction or discontinuation of SV due to these adverse effects.

**Data availability statement:** All relevant data are within the paper and its Supporting Information files.

**Funding:** The author(s) received no specific funding for this work.

**Competing interests:** The authors have declared that no competing interests exist.

## Conclusions

In MHD patients with hypertension, SV demonstrates a preliminary ability to lower blood pressure and appears safe. Further prospective randomized studies are warranted to validate our findings.

## Introduction

Hypertension is the most prevalent complication among patients with chronic kidney disease (CKD), affecting more than 80% of individuals undergoing maintenance hemodialysis (MHD).[1] Among these MHD patients, hypertension is frequently severe and significantly contributes to the elevated rates of cardiovascular events and mortality.[2] A meta-analysis of 30 randomized clinical trials, which included over 15,000 patients, demonstrated that more aggressive hypertension management is associated with reduced mortality rates in patients with CKD who also have hypertension.[3] Furthermore, studies have demonstrated that the beneficial effects of blood pressure (BP) control on mortality are particularly pronounced in patients with end-stage renal disease (ESRD).[4,5]

Sacubitril-valsartan (SV) is the world's first marketed angiotensin receptor neprilysin inhibitor. It can simultaneously regulate the renin–angiotensin–aldosterone system (RAAS) and the natriuretic peptide system, and is now used to treat chronic heart failure (HF) as well as primary hypertension. Compared with angiotensin-converting enzyme inhibitors (ACEIs), the use of SV has been shown to reduce both mortality and hospitalization for heart failure in a symptomatic patient population with a reduced ejection fraction (EF).[6] Moreover, SV can provide additional clinical benefits independent of their original purpose, such as alleviating of renal impairment in patients with HF.[7] In particular, SV has demonstrated a significant anti-hypertensive effect and is more effective than angiotensin receptor blockers (ARBs), with few, if any, patients with ESRD and no MHD patients included in these trials.[8,9] Recently, several preliminary studies have investigated the application of SV in patients with advanced CKD (stage IV or V) and HF, yielding positive results characterized by lower overall mortality, cardiovascular death, and rates of hospital readmissions. [10–12] Furthermore, Ding Y et al. suggested that SV may play a cardioprotective role in patients with ESRD undergoing either hemodialysis or peritoneal dialysis.[13] Despite these innovative work, data on MHD patients with hypertension have not been previously explored. Specifically, the efficacy and safety of SV in this population remain unknown. Therefore, to address these gaps, we conducted this study to investigate the efficacy and safety of SV in MHD patients with hypertension.

## Methods

### Participants

We conducted a retrospective analysis of data from patients undergoing MHD at Youyang Hospital, First Affiliated Hospital of Chongqing Medical University, from January 1, 2023 to June 30, 2024. Data was collected between October 1, 2024 and

December 20, 2024. The decision to initiate SV treatment was made following discussions between the patient and their nephrologist and/or cardiologist. Patients received SV treatment in conjunction with conventional therapy, which included regular hemodialysis (2–3 times per week, 4 hours per session), anti-hypertensives (such as β-blockers, α-blockers, and calcium channel blockers), agents to improve anemia (including iron supplements and recombinant human erythropoietin), mineral homeostasis regulation, and glycemic control for diabetic patients. These treatments were generally maintained at stable dosages during the follow-up period, unless clinical circumstances (e.g., significant hypotension, new adverse events, or a clear clinical need for adjustment) explicitly necessitated a change. At the beginning of SV treatment, the initial anti-hypertensive medications were either directly replaced with SV, or the ACEIs/ARBs in the original anti-hypertensive regimens were replaced with SV. Alternatively, SV can be added to existing anti-hypertensive regimens to achieve better BP control. On hemodialysis days, SV was generally administered after the dialysis session to mitigate the risk of intradialytic hypotension. The initial dose of SV (Beijing Novartis Pharmaceuticals Co., Ltd.; approval number: State Drug License J20171054) was 50 mg, and it was administered twice daily. The follow-up dose was adjusted to between 50 mg twice daily to 200 mg twice daily on the basis of patient tolerance.

The inclusion criteria for this study were: (1) aged ≥18 years with a diagnosis of hypertension, and (2) underwent MHD due to ESRD for more than 3 months. The exclusion criteria included: (1) systolic BP (SBP) below 90 mmHg or diastolic BP (DBP) below 60 mmHg; (2) hyperkalemia (serum potassium of at least 5.5 mmol/L); (3) known allergies to SV; (4) poor compliance, defined as an inability to cooperate with medication and examination; (5) irregular hemodialysis sessions and evident hypervolemia; and (6) severe hepatic dysfunction, biliary cirrhosis, or cholestasis.

The ethical approval of the study was obtained from the Ethics Committee of Youyang Hospital, First Affiliated Hospital of Chongqing Medical University (2024YYXRMYY04), and was conducted according to the principles of the Helsinki Declaration.[14] In view of the observational and retrospective study design, the ethics committee exempted the need for informed consent.

## Data collection

Demographic and clinical information was gathered within one week prior to or following the initiation of SV. The data included age, gender, body mass index (BMI), and the main causes of ESRD, along with comorbidities such as cardiovascular disease, diabetes, cerebrovascular disorders, and peripheral vascular conditions. Additionally, laboratory results were documented, which included levels of serum platelet (PLT), hemoglobin (HGB), alanine aminotransferase (ALT), aspartate aminotransferase (AST), albumin, serum calcium, serum phosphorus, serum potassium, and parathyroid hormone levels.

## BP monitoring

The patient's predialysis BP was recorded by a nurse at the dialysis center via a validated automated oscillometric device (HEM-7132,OMRON Healthcare, Kyoto, Japan). The predialysis BP measurements were obtained following a standardized protocol in accordance with the American Heart Association (AHA) guidelines to ensure accuracy and consistency.[15] Measurements were taken in a quiet, temperature-controlled environment after the participant had been seated comfortably with back support and feet flat on the floor for a minimum of 5 minutes. For the measurement process, the arm selected for the measurement was the one opposite to the arm with a functioning dialysis access.This chosen arm was carefully positioned and supported at the level of the heart, and an appropriately sized cuff (bladder encircling ≥80% of the arm circumference) was applied to the bare upper arm.[16] The baseline BP was established by calculating the mean of 6 consecutive predialysis BP measurements. Pulse pressure was calculated as the difference between SBP and DBP. The primary intervention for SBP management involves meticulous dry weight adjustment through careful clinical assessment. If patients exhibited symptoms such as muscle cramps, excessive saline needs, or symptomatic hypotension, and could not tolerate ultrafiltration, supplemental anti-hypertensive medications were systematically prescribed according to

established treatment protocols.[16] After SV was used for 3 months, the mean of 6 consecutive measurements of predialysis BP was recorded for analysis. The requirements for BP measurement remained consistent with previous protocols.

## Safety and efficacy outcomes

The efficacy outcomes were the changes in the mean values of SBP, DBP, and pulse pressure from baseline to three months after SV treatment. The safety outcomes of the study were new-onset intradialytic hypotension, cough, angioedema (characterized by subcutaneous and/or submucosal swelling), hyperkalemia (serum potassium of at least 5.5 mmol/L) and abnormal AST/ALT. Intradialytic hypotension was defined as a reduction in SBP of ≥ 20 mmHg or mean BP of ≥ 10 mmHg during dialysis treatment, accompanied by symptoms such as cramping, headache, lightheadedness, vomiting, chest pain, or necessitating intervention (including reduction in ultrafiltration, administration of fluids, or reduction of blood pump flow).

## Statistical analysis

All statistical analyses and diagrams via GraphPad Prism, version 9.0, and SPSS version 27.0 software (IBM Corp., Armonk, NY). Categorical variables are presented as percentages (%) and counts, whereas normally distributed continuous variables are reported as the means±standard deviations. Nonnormally distributed continuous variables are represented by the median and interquartile range. To compare the mean values before and after treatment, a paired-samples t test was used if the differences followed a normal distribution; conversely, the Wilcoxon signed-rank test for paired samples was utilized when the differences did not adhere to a normal distribution. P<0.05 was considered to be statistically significant.

## Results

### Baseline data

This study initially screened 206 patients with MHD at our hospital were included from January 2023 to June 2024, among whom a total of 71 patients were identified with hypertension. According to the exclusion criteria, 5 patients with hemodialysis less than three mouths and 2 patients with poor compliance were excluded from this study, and 64 patients were ultimately included (Fig 1). Among them, 15 patients transitioned to SV after discontinuing ACEI/ARB

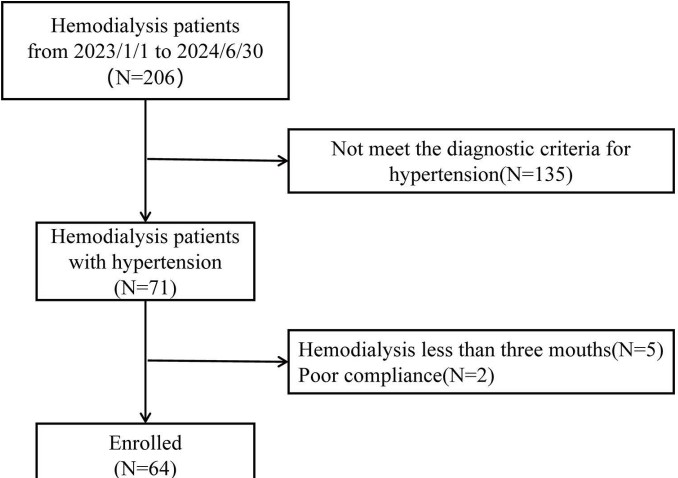

**Fig 1. Flow diagram for inclusion and analysis of maintenance hemodialysis with hypertension.**

medications. The characteristics at baseline for the MHD patients with hypertension are presented in Table 1. The mean age was 54.11 ± 13.78 years, and 47 (73.4%) were males. The average BMI was 22.30 ± 3.30, and the median dialysis age was 49 years (20.75–74.75). The primary reason for ESRD was diabetes-related kidney disease (34.3%), followed by glomerulonephritis (21.2%), hypertensive nephropathy (23.4%), and others (10.9%). Dialysis was conducted three times a week for 56 patients (87.5%) and twice a week for 8 patients (12.5%). With respect to other medications, 45 patients (70.3%) were prescribed calcium channel blockers, 10 (15.6%) were on β-blockers, and 20 (31.2%) were treated with α-blockers.

Among these patients, treatment commenced with an initial dosage of 50 mg administered twice daily, with titration occurring every 2–4 weeks up to a maximum of 200 mg twice daily. All patients maintained SV treatment throughout the study period. The final dosage of SV in the study group is detailed in Table 2. The numbers of MHD patients with hypertension who received 50 mg twice daily, 100 mg twice daily, and 200 mg twice daily were 11 (17.19%), 38 (59.38%), and 15 (23.44%), respectively.

**Table 1. Baseline data of patients in maintenance hemodialysis with hypertension.**

| Variables | N = 64 |
|---|---|
| Age | 54.11 ± 13.78 |
| Male gender, n (%) | 47 (73.4) |
| BMI, (kg/m²) | 22.30 ± 3.30 |
| Dialysis age, (months) | 49.00 (20.75-74.75) |
| Causes of ESRD | |
| Glomerulonephritis, n (%) | 20 (31.2) |
| Diabetes related kidney disease, n (%) | 22 (34.3) |
| Hypertensive nephropathy, n (%) | 15 (23.4) |
| Others, n (%) | 7 (10.9) |
| Frequency of dialysis | |
| 3 times per week, n (%) | 56 (87.5) |
| 2 times per week, n (%) | 8 (12.5) |
| Other medications | |
| calcium channel blocker, n (%) | 45 (70.3) |
| β-blocker, n (%) | 10 (15.6) |
| α-blocker, n (%) | 20 (31.2) |
| Laboratory indices | |
| blood urea nitrogen, (mmol/L) | 20.15 (17.40–25.55) |
| serum creatinine, (μmol/L) | 982.00 (804.00–1275.10) |
| serum uric acid, (μmol/L) | 455.03 ± 86.03 |
| Serum calcium, (mmol/L) | 2.13 (2.03–2.24) |
| Serum phosphate, (mmol/L) | 1.63 (1.23–2.23) |
| Ferritin, (ng/ml) | 40.30 (25.50–87.00) |
| Serum potassium, (mmol/L) | 4.86 ± 0.99 |
| Hemoglobin, (g/L) | 113.00 (95.00–123.00) |
| Parathyroid hormone, (pg/ml) | 242.00 (151.00–449.00) |
| Albumin, (g/L) | 37.10 (33.30–42.00) |

Continuous values are documented by Mean±SD or Median (interquartile); Categorical variables are recorded by n (percentage). BMI, body mass index;ESRD, end-stage renal disease.

## Comparison of BP before and after SV treatment

The comparisons of BP before and after the administration of SV are presented in Table 3 and Fig 2. At baseline, the mean SBP prior to the administration of SV was 163.13±15.10 mmHg, the mean DBP was 92.03±13.88 mmHg, and the mean pulse pressure was 71.13±14.18 mmHg. After 3 months of treatment with SV, the mean SBP, DBP, and pulse pressure were 142.06±13.85 mmHg, 83.59±10.12 mmHg, and 58.47±13.34 mmHg, respectively. Furthermore, the mean decreases in SBP, DBP, and pulse pressure were 21.06±15.82 mmHg (P<0.001), 8.41±13.09 mmHg (P<0.001)

**Table 2. Final dose of sacubitril/valsartan in maintenance hemodialysis with hypertension.**

| Final dose | Number of patients (%) |
|---|---|
| 50 mg twice daily | 11/64 (17.19) |
| 100 mg twice daily | 38/64 (59.38) |
| 200 mg twice daily | 15/64 (23.44) |

**Table 3. Comparison of blood pressure changes before and after sacubitril-valsartan administration of patients in maintenance hemodialysis with hypertension.**

| Blood pressure | Baseline | Follow-up | P | The degree of blood pressure drop | 95%CI |
|---|---|---|---|---|---|
| Systolic pressure before dialysis (mmHg) | 163.13±15.10 | 142.06±13.85 | <0.001 | 21.06±15.82 | 17.11–25.02 |
| Diastolic pressure before dialysis (mmHg) | 92.03±13.88 | 83.59±10.12 | <0.001 | 8.41±13.09 | 5.14–11.68 |
| Pulse pressure (mmHg) | 71.13±14.18 | 58.47±13.34 | <0.001 | 12.65±13.17 | 0.36–15.94 |

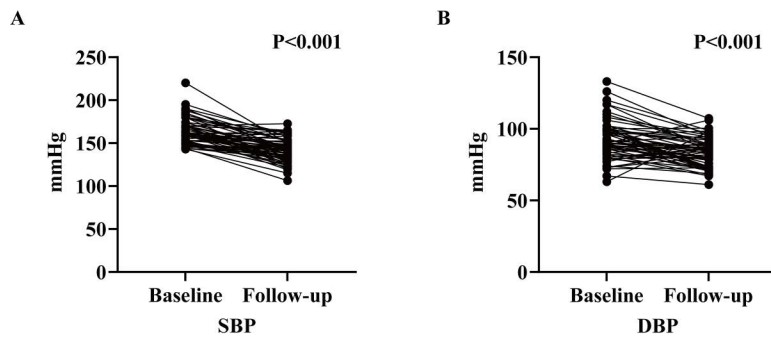

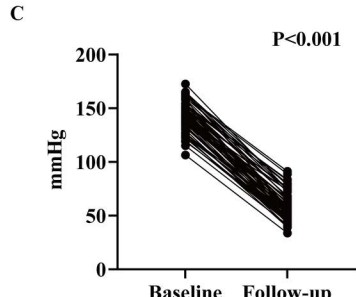

**Fig 2. Comparison of blood pressure changes before and after sacubitril/valsartan administration in maintenance hemodialysis with hypertension.**

**Table 4. Adverse events of patients in maintenance hemodialysis with hypertension.**

| Adverse events | sacubitril-valsartan |
|---|---|
| New-onset intradialytic hypotension | 3/64 (4.69%) |
| Hyperkalemia (>5.5 mmol/L) | 12/64 (18.75%) |
| Cough | 0/64 (0%) |
| Angio-edema | 0/64 (0%) |
| Abnormal AST/ALT (>40 U/L) | 0/64 (0%) |

ALT, alanine aminotransferase; AST, aspartate aminotransferase.

and $12.65 \pm 13.17$ mmHg (P < 0.001), respectively. In summary, these findings indicatet that SV treatment improves BP in patients receiving MHD with hypertension.

### Drug safety evaluation

The adverse events are summarized in Table 4. New-onset intradialytic hypotension was recorded in three patients. This study involved twelve patients with hyperkalemia. No patients reported experiencing cough, angioedema, or abnormal liver biochemistry. Among the 64 research subjects included in the study, no patients underwent dose reduction or discontinuation of SV due to these adverse effects.

### Discussion

This research investigated the efficacy and safety of SV in MHD patients with hypertension. Our findings demonstrate that SV treatment is effective for controlling BP in these patients, as evidenced by a reduction in the mean SBP, DBP, and pulse pressure. Furthermore, SV treatment has shown a favorable safety profile in patients undergoing MHD, with no instances of dose reduction or discontinuation of SV due to adverse effects.

Hypertension, a significant and modifiable risk factor for cardiovascular disease, is observed in over 80% of patients receiving MHD.[17, 18] In individuals who do not adequately control fluid levels, an increase in cardiac output resulting from excess volume or sodium can lead to hypertension.[19–21] Furthermore, patients undergoing MHD exhibit increased arterial stiffness, which is partially attributed to heightened inflammation, reduced renal excretion of vascular toxins, and maladaptive metabolic and hormonal processes, ultimately contributing to premature vascular aging.[22] Additionally, patients undergoing MHD frequently exhibit sustained sympathetic overactivity, a dysregulated mechanism that is exacerbated by fluid accumulation and arterial stiffness.[23] The overactivity of the RAAS is demonstrated through elevated serum renin and aldosterone concentrations during pre- and post-dialysis phases, potentially as a compensatory reaction to ultrafiltration-mediated redistribution of sodium balance and fluid dynamics.[24] In summary, the key drivers of elevated BP among MHD recipients include fluid accumulation, arterial stiffness, hyperactivity of the sympathetic nervous system, activation of the RAAS, endothelial dysfunction, and the administration of erythropoietin-stimulating agents.[19,20] Treatment with anti-hypertensive agents can significantly reduce cardiovascular morbidity, cardiovascular mortality, and overall mortality in dialysis patients.[25] Hypertension in MHD patients presents several unique diagnostic, prognostic and therapeutic challenges.[26] Hypertension serves as both a contributing factor and consequence in CKD progression, with widespread impact on affected individuals. The therapeutic management of BP in patients undergoing MHD continues to pose significant clinical challenges.

SV exerts a twofold effect on the primary regulators of the cardiovascular system, specifically the RAAS and the natriuretic peptide (NP) system. Beyond its clinically encouraging advantage of lowering cardiovascular mortality and hospitalization for HF in patients with HF, especially in those with reduced EF.[6,27,28] The exceptional ability of SV to reduce BP has been consistently demonstrated in various clinical studies focusing on hypertension.[29] Moreover, increasing

evidence suggests that SV surpasses traditional RAS blockers in its effectiveness at lowering BP in individuals with hypertension.[7] Additionally, SV may be effective in treating apparent resistant hypertension.[30] Elevated blood pressure, as a major adjustable contributor to cardiac remodeling and HF.[31] The clinical experience with SV in CKD primarily pertains to patients with CKD who also have HF.[11,32–34] The UK HARP-III trial evaluated the efficacy of SV and irbesartan in treating 414 cases of HF complicated with CKD, SV has an additional effect of reducing BP in patients with CKD.[35] A 12-week, single-center, prospective, before-and-after study indicated that SV effectively and safely managed resistant hypertension while improving myocardial workload and quality of life in hemodialysis patients. Nonetheless, the study's limitations include a small sample size, with only 18 patients ultimately enrolled and completing the study.[36] Our study yielded similar results; however, we included a larger sample size.

Compared to ACEIs, SV has demonstrated efficacy and safety in preventing cardiovascular morbidity and overall mortality among patients with HF. However, ESRD patients were not included in the randomized controlled trial.[37] Its application remains controversial, particularly due to the potential risks of hypotension and hyperkalemia. Data on the use of the SV in patients undergoing MHD are scarce. Studies have demonstrated that SV therapy is associated with elevated serum potassium levels in anuric hemodialysis patients; however, the proportion of patients experiencing severe hyperkalemia did not exhibit a significant increase.[38,39] These findings suggest that the application of SV in patients undergoing MHD is relatively safe. Among the 64 research subjects included in the study, we did not impose a strict prohibition on patients with when hyperkalemia regarding the use of SV. Hyperkalemia was managed through a combination of adequate dialysis and the administration of potassium-lowering medications. Notably, patients with hyperkalemia did not experience further deterioration in blood potassium levels or complications associated with the condition. Additionally, none of the patients experienced coughing or angioedema. Three patients experienced symptomatic hypotension; however, this was corrected through volume adjustment and did not necessitate a reduction or discontinuation of SV. Furthermore, there was no statistically significant difference in ALT or AST levels among all patients before and after medication. Overall, MHD patients demonstrated good tolerance to SV.

This study has several limitations that should be acknowledged. Firstly, as a single-center retrospective investigation, which introduces potential selection bias and numerous confounding factors. Fifteen patients transitioned directly from ACEI/ARB to SV without a washout period. Although this approach is common in clinical practice, it may lead to a carry-over effect from previous medications. Secondly, the follow-up period of this study was relatively short, necessitating further observation of the subsequent survival outcomes of the patients. Therefore, larger prospective, randomized controlled trials (including phase 3 clinical trials) are necessary to further evaluate the efficacy and safety of SV. Thirdly, predialysis BP measurements may be less effective than home BP monitoring (HBPM) and ambulatory BP monitoring (ABPM) in predicting clinical outcomes.[40,41] HBPM and ABPM offer a more comprehensive and continuous assessment of BP, capturing fluctuations that may be missed by predialysis measurements. Nevertheless, the implementation of HBPM and ABPM is often hampered by an increased financial burden on patients and poorer adherence. These practical challenges need to be considered when evaluating the utility of different BP measurement methods in clinical research and practice. Finally, in this study dry weight assessment was performed only by using clinical indices. However, we know that subclinical hypervolemia is common among patients with MHD. So, a more objective method of assessing dry weight should be used (bioelectricalimpedance analysis, lung ultrasound, inferior vena cava diameter).

## Conclusion

In conclusion, this study has preliminarily investigated the efficacy and safety of SV in individuals with MHD and hypertension. The results indicate its potential applicability within this population. However, to further validate and confirm the specific clinical value of SV in this particular patient group, future studies should be rigorously designed, larger in scale, and include randomized controlled trials (including phase 3 clinical trials).

## Supporting information

**S1 File. STROBE statement.**
(DOCX)

**S2 File. S Dataset.**
(XLSX)

## Author contributions

**Conceptualization:** Lili Jiang, Jurong Yang, Shaofa Wu.

**Data curation:** Lili Jiang, Jing Ran, Yu Zhu, Lupin Pan, Bayi Yang, Xue Ran, Ying Ran, Hejun Ding.

**Methodology:** Lili Jiang, Jurong Yang.

**Project administration:** Lili Jiang.

**Writing – original draft:** Lili Jiang, Jurong Yang, Shaofa Wu.

**Writing – review & editing:** Lili Jiang, Jurong Yang, Shaofa Wu.

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
