## [Decision Letter · Decision Letter 0]

16 Jun 2025

Dear Dr. Wu,

We look forward to receiving your revised manuscript.

Kind regards,

Hean Teik Ong

Academic Editor

PLOS ONE

Journal Requirements:

Additional Editor Comments:

Please make major revision to address comments from the reviewers.

Reviewers' comments:

Reviewer's Responses to Questions

**Comments to the Author**

1. Is the manuscript technically sound, and do the data support the conclusions?

Reviewer #1: No

Reviewer #2: Yes

2. Has the statistical analysis been performed appropriately and rigorously?

Reviewer #1: I Don't Know

Reviewer #2: Yes

3. Have the authors made all data underlying the findings in their manuscript fully available?

Reviewer #1: Yes

Reviewer #2: Yes

4. Is the manuscript presented in an intelligible fashion and written in standard English?

Reviewer #1: Yes

Reviewer #2: Yes

Reviewer #1: This study evaluated efficacy and safty of sacubitril/valsartan in patients with maintanance HD.

I have some concerns for this article.

The primary outcome of the study needs design of phase 3 clinical trial.

I am not so sure that retrospective study can give proper strength to conclusions of this study, especially efficacy.

There are too many variables that can affect the efficacy in retrospecitve data and it is necessary that the inclusion criteria should be more specific and study should be conduction in highly selected patients group.

The inclusion criteria of this study is somewhat broad and non-specific.

Authors addressed 15 patients were included who switched medication from ACEI/ARB to sacubitril/varsartan, these patients should be excluded or need at least 2 weeks of washout periods in phase 3 clinical trials.

As some patients were treated with calcium channel blocker and alpha, beta blockers and some were not, but it is not clearly described that these medications were fixed (dosage and/or combination) during 3 months.

As the volume status can affect the blood pressure in maintanance HD patients, it is necessary to decribe the dry weight or other parameters of HD during 3 months period were fix or not.

As this is retrospective study, maybe it would be better to compare efficacy in patients with ACEI/ARB and sacubitril/varsartan.

Reviewer #2: Overall this manuscript presented in a technically sound scientific research. In regards on the study product of Sacubitril-valsartan. It is well known it can cause hyperkalaemia. End stage renal failure patient populations are already prone to be hyperkalaemic. Thus Exclusion Criteria may highlight excluding those existing hyperkalaemic patients. for patients on maintenance hemodialysis, antihypertensive may not be routinely served prior to the hemodialysis session as patients are prone for intradialytic hypotension. Thus emphasizing timing of BD dosing Sacubitril-valsartan on hemodialysis day could be crucial. Either it is omitted prior hemodialysis session and taken only after hemodialysis was not stated clearly in the study. However overall the study is well written

**Do you want your identity to be public for this peer review?** For information about this choice, including consent withdrawal, please see our Privacy Policy

Reviewer #1: No

Reviewer #2: No

---

## [Author Response · Author response to Decision Letter 1]

5 Jul 2025

Dear journal editors and reviewers:

Thank you for your careful reading and kind reminder of my manuscript.I admire your rigorous logic and pragmatic academic spirit.Following is my responds to each point raised.

Journal Requirements:

Comment: 1.Please ensure that your manuscript meets PLOS ONE's style requirements, including those for file naming.

Response: Thank you very much for your advice. The manuscript has been reformatted to meet the PLOS ONE’s style requirements.

Comment: 2.We note that the grant information you provided in the ‘Funding Information’ and ‘Financial Disclosure’ sections do not match. When you resubmit, please ensure that you provide the correct grant numbers for the awards you received for your study in the ‘Funding Information’ section.

Response: Thank you for your comments. We have carefully checked and corrected the grant information to ensure that the ‘Funding Information’ and ‘Financial Disclosure’ sections are now identical and accurate. The correct grant number is 2024YY760008.

Comment: 3. We note that your Data Availability Statement is currently as follows: [All relevant data are within the manuscript and its Supporting Information files. ] Please confirm at this time whether or not your submission contains all raw data required to replicate the results of your study.

Response: Thank you for your comments. We acknowledge that our previous statement was insufficient. To ensure full transparency and reproducibility, we have compiled the minimal dataset underlying our findings (including values behind means, standard deviations, and data used for figures) and have prepared it to be uploaded as a Supporting Information file (S Data.xlsx).

Comment: 4. Please amend either the abstract on the online submission form (via Edit Submission) or the abstract in the manuscript so that they are identical.

Response: Thank you for your comments. We have ensured that the abstract in the manuscript is identical to the one entered in the online submission form.

Comment: 5. Please include captions for your Supporting Information files at the end of your manuscript, and update any in-text citations to match accordingly.

Response: Thank you for your comments. We have now added a Supporting Information file (S Data.xlsx) containing the minimal dataset. We have also included a corresponding caption for this file at the end of the manuscript, and updated any in-text references as needed.

Modification:

Additional Editor Comments:

Comment: Please make major revision to address comments from the reviewers.

Response: : Thank you for your comments. We have thoroughly reviewed all reviewer comments and made comprehensive revisions to address each point.

Reviewers' comments:

Reviewer #1:

Comment 1: The primary outcome of the study needs design of phase 3 clinical trial. I am not so sure that retrospective study can give proper strength to conclusions of this study, especially efficacy.

Response: Thank you for your comments. We fully agree with your perspective. Its primary purpose was to provide exploratory evidence of the preliminary efficacy and safety of sacubitril-valsartan in a specific patient population (maintenance hemodialysis patients with hypertension) where data is currently limited. In the context of this data scarcity, our study aims to fill a gap and offer insights and foundations for future research directions. We fully acknowledge the inherent limitations of retrospective studies. Therefore, the findings of this study should be regarded as preliminary results, intended to lay the groundwork and guide future larger-scale, more rigorously designed prospective randomized controlled trials (including phase 3 clinical trials) to further validate and confirm the precise efficacy and safety of sacubitril-valsartan in this population. We have revised the language in the abstract and conclusion to more appropriately frame our findings as preliminary observations, and to strongly emphasize the necessity of future prospective, randomized trials.(Page 3, Lines 57-58; Page 17, Lines 351-357)

Comment 2: There are too many variables that can affect the efficacy in retrospective data and it is necessary that the inclusion criteria should be more specific and study should be conduction in highly selected patients group. The inclusion criteria of this study is somewhat broad and non-specific.

Response: Thank you for your comments. We agree that numerous variables in retrospective data can influence treatment outcomes, and stricter inclusion criteria would improve the homogeneity and specificity of the study population. As a real-world, retrospective study, our aim was to reflect routine clinical practice and explore the effectiveness of SV in a broader population of MHD patients with hypertension, for whom current data is severely limited. We did implement specific exclusion criteria to mitigate some confounding factors, such as patients with poor compliance, irregular hemodialysis sessions, and evident hypervolemia, which are critical for BP management in this population. While a highly selected patient group is indeed ideal for a prospective trial, our study provides valuable preliminary insights from a more general clinical setting. We deeply appreciate your good suggestion. We have added the explanation in the revised manuscript.(Page 16, Lines 327-328)

Comment 3: Authors addressed 15 patients were included who switched medication from ACEI/ARB to sacubitril/varsartan, these patients should be excluded or need at least 2 weeks of washout periods in phase 3 clinical trials.

Response: Thank you for your comments. We completely agree with the reviewer that in a prospective phase 3 clinical trial, a washout period for ACEI/ARB would be essential to isolate the effect of SV. However, this study is retrospective and reflects real-world clinical practice where direct switches often occur without a washout due to immediate clinical needs and patient convenience. We acknowledge that the prior use of ACEI/ARB in 15 patients could introduce a carry-over effect, potentially influencing the observed blood pressure reductions attributed to SV. We have explicitly added this as a limitation in the revised manuscript.(Page 16, Lines 328-331 )

Comment 4: As some patients were treated with calcium channel blocker and alpha, beta blockers and some were not, but it is not clearly described that these medications were fixed (dosage and/or combination) during 3 months.

Response: Thank you for your comments. We appreciate this important point regarding concomitant medications. In this study, while all patients continued their other anti-hypertensive medications, the specific dosages and combinations of calcium channel blockers, alpha blockers and beta blockers were generally maintained stable throughout the 3-month follow-up period, unless clinical circumstances (e.g., significant hypotension, new adverse events, or a clear clinical need for adjustment) explicitly necessitated a change. However, we acknowledge that detailed records on precise dosage adjustments for these co-medications were not consistently available for every patient in the retrospective data, which represents a limitation. We have clarified this in the Methods section to emphasize that these medications were generally stable.(Page 6, Lines 110-113)

Comment 5: As the volume status can affect the blood pressure in maintanance HD patients, it is necessary to decribe the dry weight or other parameters of HD during 3 months period were fix or not.

Response: Thank you for your comments. We strongly agree with the reviewer on the critical importance of volume status in hypertension management for MHD patients. Our study primarily relied on clinical assessment for dry weight adjustment. While efforts were made to achieve and maintain optimal dry weight for each patient, detailed objective parameters (e.g., bioelectrical impedance analysis, lung ultrasound, inferior vena cava diameter) were not systematically available in our retrospective dataset. This represents a notable limitation of our study. (Page 16-17, Lines 344-349)

Comment 6: As this is retrospective study, maybe it would be better to compare efficacy in patients with ACEI/ARB and sacubitril/varsartan.

Response: Thank you for your comments. We appreciate this valuable suggestion. Comparing the efficacy of SV directly with ACEI/ARB in a matched cohort of MHD patients with hypertension would undoubtedly provide more robust evidence. However, due to the retrospective, single-arm nature of our study design and the lack of a suitable control group with comparable baseline characteristics and treatment protocols within our retrospective data, such a direct comparison was not feasible in the current analysis.

Reviewer #2:

Comment 1: Overall this manuscript presented in a technically sound scientific research. In regards on the study product of Sacubitril-valsartan. It is well known it can cause hyperkalaemia. End stage renal failure patient populations are already prone to be hyperkalaemic. Thus Exclusion Criteria may highlight excluding those existing hyperkalaemic patients.

Response: Thank you for your constructive comments. Our exclusion criterion for hyperkalemia was defined as a serum potassium level of at least 5.5 mmol/L. This threshold was deliberately chosen to permit the inclusion of patients with serum potassium levels ranging from 5.0 to 5.5 mmol/L. This decision reflects the evolving understanding that mild hyperkalemia should not always be an absolute contraindication for the use of renin-angiotensin-aldosterone system inhibitors such as SV, especially given their critical cardiorenal protective benefits in high-risk populations, including patients on maintenance hemodialysis. By including such patients, our retrospective study aims to represent real-world clinical practice, where hyperkalemia is a prevalent and manageable issue in maintenance hemodialysis patients. We have clarified this exclusion criterion in the revised manuscript to enhance readers' comprehension of our patient selection process.(Page 7, Lines 127-128)

Comment 2: For patients on maintenance hemodialysis, antihypertensive may not be routinely served prior to the hemodialysis session as patients are prone for intradialytic hypotension. Thus emphasizing timing of BD dosing Sacubitril-valsartan on hemodialysis day could be crucial. Either it is omitted prior hemodialysis session and taken only after hemodialysis was not stated clearly in the study.

Response: Thank you very much for your advice. This is a crucial practical point for maintenance hemodialysis patient, given the inherent risk of intradialytic hypotension. In our center's standard clinical practice, for patients receiving sacubitril-valsartan on hemodialysis days, the medication is typically administered after the hemodialysis session. This strategy aims to minimize the risk of severe intradialytic hypotension. We have clarified this point in the revised discussion (See Page 6 Line 118-119).

---

## [Decision Letter · Decision Letter 1]

25 Jul 2025

Efficacy and safety of sacubitril-valsartan in maintenance hemodialysis patients with hypertension:A retrospective study

PONE-D-25-09976R1

Dear Dr. Shaofa Wu,

We’re pleased to inform you that your manuscript has been judged scientifically suitable for publication and will be formally accepted for publication once it meets all outstanding technical requirements.

Thank you for submitting your interesting article to PLOS ONE.  Your article has been accepted by the reviewers and academic editor.

You will now have to communicate with the administrative editors about meeting the formatting requirements of the journal.

Kind regards,

Hean Teik Ong, FRCP, FACC

Academic Editor

PLOS ONE

Additional Editor Comments (optional):

Thank you for submitting your interesting article to PLOS ONE. Your article has been accepted by the reviewers and academic editor.

You will now have to communicate with the administrative editors about meeting the formatting requirements of the journal.

Reviewers' comments:

Reviewer's Responses to Questions

**Comments to the Author**

Reviewer #1: All comments have been addressed

Reviewer #2: All comments have been addressed

2. Is the manuscript technically sound, and do the data support the conclusions?

Reviewer #1: Yes

Reviewer #2: Yes

3. Has the statistical analysis been performed appropriately and rigorously?

Reviewer #1: Yes

Reviewer #2: I Don't Know

4. Have the authors made all data underlying the findings in their manuscript fully available?

Reviewer #1: Yes

Reviewer #2: Yes

5. Is the manuscript presented in an intelligible fashion and written in standard English?

Reviewer #1: Yes

Reviewer #2: Yes

Reviewer #1: Authors responsed to all comments from previous review.

I don't have furthor comments.

Reviewer #2: My concerns and comments had been well explained. Even though this is a retrospective study, the study was well designed I felt is relevant in my field to look at the safety of usage of sacubitril-valsartan in hemodialysis patients

**Do you want your identity to be public for this peer review?** For information about this choice, including consent withdrawal, please see our Privacy Policy

Reviewer #1: No

Reviewer #2: **Yes: ** Zhang Duan Tham

---

## [Editor Report · Acceptance letter]

PONE-D-25-09976R1

PLOS ONE

Dear Dr. Wu,

I'm pleased to inform you that your manuscript has been deemed suitable for publication in PLOS ONE. Congratulations! Your manuscript is now being handed over to our production team.

Kind regards,

on behalf of

Dr. Hean Teik Ong

Academic Editor

PLOS ONE